# Spontaneously formed phonon frequency combs in van der Waals solid CrGeTe$_3$ and CrSiTe$_3$

Lebing Chen [1,2,11] ✉, Gaihua Ye [3,11], Cynthia Nnokwe[3], Xing-Chen Pan[4], Katsumi Tanigaki[4,5], Guanghui Cheng[4,6,7], Yong P. Chen [4,6,7,8], Jiaqiang Yan [9,10], David G. Mandrus [9,10], Andres E. Llacsahuanga Allcca[6], Nathan Giles-Donovan[1,2], Robert J. Birgeneau [1,2] ✉ & Rui He [3] ✉

Optical phonon engineering through nonlinear effects has been utilized in ultrafast control of material properties. However, nonlinear optical phonons typically exhibit rapid decay due to strong mode-mode couplings, limiting their effectiveness in temperature or frequency sensitive applications. Here we report the observation of long-lived nonlinear optical phonons through the spontaneous formation of phonon frequency combs in the van der Waals material CrXTe$_3$ (X=Ge, Si) using high-resolution Raman scattering. Unlike conventional optical phonons, the highest $A_g$ mode in CrGeTe$_3$ splits into equidistant, sharp peaks forming a frequency comb that persists for hundreds of oscillations and survives up to 200K. These modes correspond to localized oscillations of Ge$_2$Te$_6$ clusters, isolated from Cr hexagons, behaving as independent quantum oscillators. Introducing a cubic nonlinear term to the harmonic oscillator model, we simulate the phonon time evolution and successfully replicate the observed comb structure. Similar frequency comb behavior is observed in CrSiTe$_3$, demonstrating the generalizability of this phenomenon. Our findings demonstrate that Raman scattering effectively probes high-frequency nonlinear phonon modes, offering insight into the generation of long-lived, tunable phonon frequency combs with potential applications in ultrafast material control and phonon-based technologies.

Optical phonon engineering with nonlinearity is extensively used to control material properties in an ultrafast manner. For instance, ultrafast light pulses can be used to achieve rapid control over magnetism, superconductivity, and ferroelectricity[1–3]. Typically, in these

processes, a femtosecond light pulse is used to excite a coherent optical phonon to a large amplitude that dynamically changes the properties of the material[4]. However, nonlinear optical phonons tend to decay rapidly due to strong mode-mode couplings, especially at

[1]Department of Physics, University of California, Berkeley, CA, USA. [2]Material Sciences Division, Lawrence Berkeley National Laboratory, Berkeley, CA, USA. [3]Department of Electrical and Computer Engineering, Texas Tech University, Lubbock, TX, USA. [4]WPI Advanced Institute for Materials Research (AIMR), Tohoku University, Sendai, Japan. [5]Department of Physics, Graduate School of Science, Tohoku University, Sendai, Japan. [6]Department of Physics and Astronomy, Purdue University, West Lafayette, IN, USA. [7]School of Electrical and Computer Engineering and Purdue Quantum Science and Engineering Institute, Purdue University, West Lafayette, IN, USA. [8]Institute of Physics and Astronomy and Villum Centers for Dirac Materials and for Hybrid Quantum Materials, Aarhus University, Aarhus-C, Denmark. [9]Department of Materials Science and Engineering, University of Tennessee, Knoxville, TN, USA. [10]Materials Science and Technology Division, Oak Ridge National Laboratory, Oak Ridge, TN, USA. [11]These authors contributed equally: Lebing Chen, Gaihua Ye. ✉e-mail: lebingchen@berkeley.edu; robertjb@berkeley.edu; rui.he@ttu.edu

higher amplitudes[5]. As a result, the tuning duration is often limited to fewer than 100 oscillation periods before the energy dissipates entirely as heat. This rapid decay presents challenges such as unwanted sample heating, which accelerates dissipation, and introduces measurement uncertainties in the frequency domain. Moreover, the availability of strong-field THz sources within the 5–15 THz (170–500 cm$^{-1}$) frequency range has been limited, restricting their capacity to fully cover the optical phonon spectrum for certain materials[6,7]. In contrast, frequency-resolved techniques such as Raman scattering and infrared spectroscopy can access phonon energies both within and beyond this range, with weaker driving amplitudes. Therefore, probing phonon nonlinearities with these techniques allow for optical phonon engineering with larger bandwidth, finer frequency resolution and greater stability against thermalization.

The decay of nonlinear optical phonons is due to anharmonic interactions in the crystal lattice, where multiple vibrational modes interact within the anharmonic atomic potential, creating numerous decay pathways that increase the likelihood of energy dissipation[8]. In the frequency domain, it will be reflected as the increased linewidth of the decaying phonon mode, as observed in bulk $Bi_2Te_3$[9,10]. Conversely, if an optical phonon is predominantly associated with a unique atomic bond and has fewer coupling pathways with other modes, it will be more difficult for this mode to decay even with anharmonicity. Instead, it may sustain parametric oscillations, leading to the formation of phonon frequency combs, while the phonon linewidth remains nearly resolution-limited.

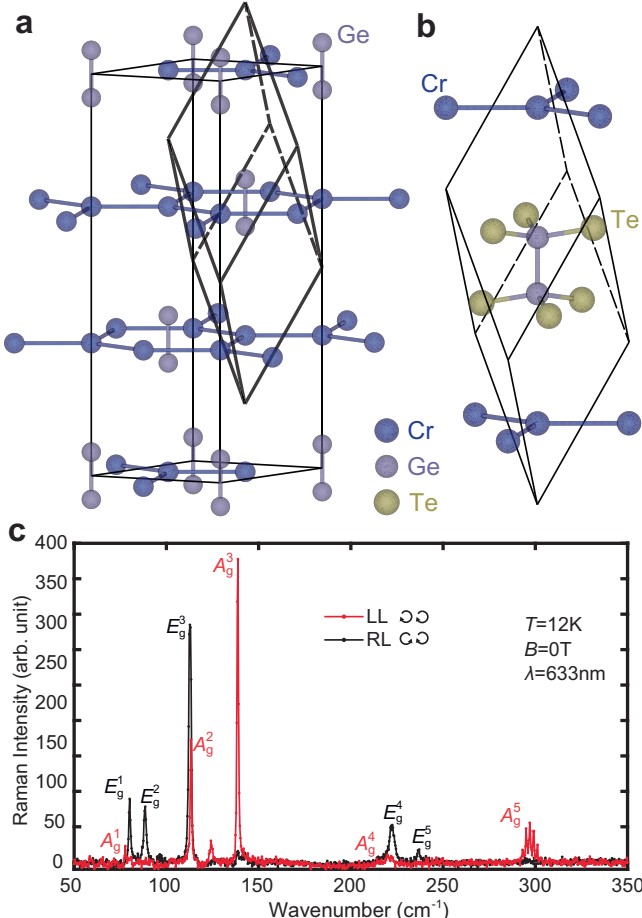

**Fig. 1 | Structure and the overall Raman spectrum of CrGrTe₃. a** Crystal structure of CrGeTe₃ in the hexagonal lattice unit. The Te atoms are not drawn. A rhombohedral lattice unit is drawn inside the hexagonal lattice unit. **b** The rhombohedral primitive cell of CrGeTe₃. **c** Overall Raman spectrum of CrGeTe₃ in parallel (LL) and cross (RL) circularly polarized channels.

The concept of frequency comb originates from optical metrology, where it is characterized by a spectrum with uniformly spaced, narrow peaks across a range of frequencies[11]. Generated from mode-locked lasers[12,13] and Kerr resonators[14], optical frequency combs are widely employed in various applications, including optical clocks[15] and frequency comb spectroscopy[16]. As similar bosonic excitations, phonon frequency combs were first described in the context of the Fermi-Pasta-Ulam-Tsingou chain, where interatomic interactions include cubic or quartic terms in addition to the quadratic Hamiltonian[17]. In this model, the population of states will oscillate for an extended period before eventually settling into thermal equilibrium[18]. Extensive research has been dedicated to generating phonon frequency combs using acoustic phonons in microresonators[19–21] or Fabry-Pérot cavities[22], where few decay paths are available. However, there are few reports on the spontaneous formation of phonon frequency combs within the optical phonon branches of crystalline solids, as most optical phonon modes typically decay.

In this study, we present the observation of spontaneously formed frequency combs in the van der Waals solid CrXTe₃ (X = Ge,Si) using high-resolution Raman scattering. The Raman spectrum of CrGeTe₃ reveals that its highest energy $A_g$ optical phonon mode splits into several equidistant, sharp peaks which persists for hundreds of oscillations and survives up to 200K. Analysis of the phonon dispersion relations and vibrations indicates that these modes exhibit flat dispersions, involving localized oscillations of $Ge_2Te_6$ clusters separated by Cr hexagons. These can effectively be regarded as independent quantum oscillators. By introducing a cubic nonlinear term into the harmonic oscillator Hamiltonian as a small perturbation, we simulate the time evolution of the phonon oscillation using both analytical and numerical tools, and successfully reproduce the comb features in the frequency domain. Similar frequency combs are observed in CrSiTe₃ as well, which can be described by the same model but with a different set of parameters.

## Results
### Raman spectrum of CrGeTe₃
The overall phonon spectrum of CrGeTe₃ from Raman scattering is relatively well-established. CrGeTe₃ belongs to the rhombohedral $R\bar{3}$ space group, where hexagonal Cr layers stack with ABC-stacking configuration, with $Ge_2Te_6$ clusters located in the center of the Cr hexagons (Fig. 1a)[23]. The primitive rhombohedral cell contains a chemical formula of $Cr_2Ge_2Te_6$ (Fig. 1b), giving rise to a total of 30 phonon modes. Among these phonon modes, there are 10 Raman active modes from $E_g^1$ to $E_g^5$ and $A_g^1$ to $A_g^5$[24]. Figure 1c shows the Raman spectra of these modes, where the incident and scattered lights are polarized in the left-hand(L) and right-hand(R) circular polarization. All 10 Raman active modes have been observed in this spectrum, with energies consistent with previous reports and ab initio calculations[24–26]. It can be shown from Raman tensor calculations that the $E_g$ modes are only visible in the cross-polarization (LR/RL) channels and the $A_g$ modes are only visible in the parallel-polarization channels (LL/RR) (see Supplementary Information Section II)[24].

The most prominent feature of the parallel-polarization Raman spectrum is the occurrence of frequency combs in the $A_g^5$ mode. As shown in detail in Fig. 2a–c, at a low temperature of 12K, the $A_g^5$ mode shows six sharp peaks instead of one broad peak as previously reported, with an equidistant spacing of 2 cm$^{-1}$ [24]. To eliminate the possibility of this being a measurement artifact, we conduct a supplementary Raman scattering experiment using a 532 nm laser (Fig. 2b) as a complement to the originally used 633 nm laser. These two experiments yield nearly identical outcomes, confirming the same positions and distances between the peaks. For both experiments, we fit the energy of the comb peaks $E_n$ as a function of their indices $n$: $E_n = E_0 - An$, with $E_0 = 305.38 \pm 0.06$ cm$^{-1}$, $A = 1.995 \pm 0.015$ cm$^{-1}$ for the 633 nm data, and $E_0 = 305.67 \pm 0.08$ cm$^{-1}$, $A = 2.017 \pm 0.021$ cm$^{-1}$ for the

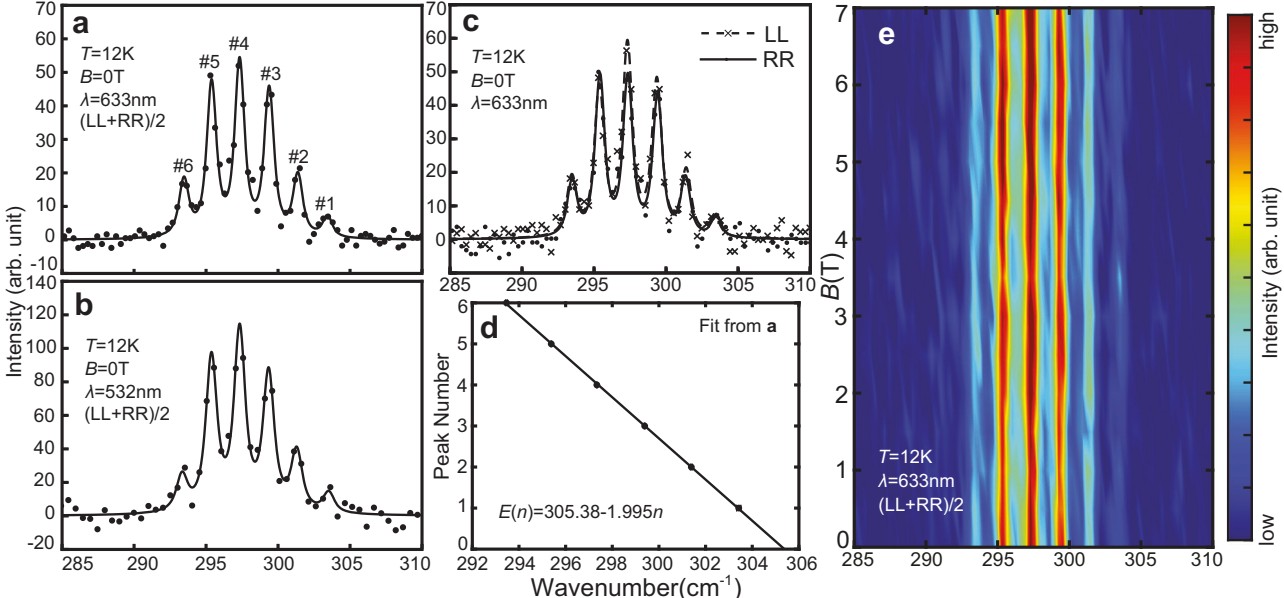

**Fig. 2 | Phonon frequency combs in CrGeTe₃ and its field dependence.**
**a, b** Detailed views of the $A_g^5$ mode of CrGeTe₃ in the parallel-polarization channel, data collected with 633 nm and 532 nm lasers respectively. All peak fits in this work are Voigt profile fits with a fixed Gaussian part equal to the instrumental resolution (see "Methods"). **c** Comparison of frequency combs between LL and RR polarization, showing similar patterns and intensities. **d** Linear fit of the comb energy as a function of the peak index. The fitting uncertainty is smaller than the size of the markers shown. **e** Magnetic field dependence of the frequency combs observed in the $A_g^5$ mode.

532 nm data. Both datasets show excellent linearity (Fig. 2d), which is one of the defining qualities of a frequency comb. Noticeably, the linewidth of these peaks are mostly resolution-limited, with minimal broadening indicating a very small amount of phonon decay.

**Field and temperature dependence**

In bulk, CrGeTe₃ is a ferromagnet below its Curie temperature $T_C$ ~65K[23,27], and it has been observed to exhibit significant magnon-phonon coupling[24–26]. This makes it crucial to investigate the potential links between the frequency combs and the spin degree of freedom, such as whether the observed splitting might be attributable to magnon scatterings in addition to phonon excitations. Two specific experiments are performed to explore the connections between magnetic order and phonon frequency combs. The first involves Raman scattering with a magnetic field applied along the c-axis. Figure 2e displays the field dependence of the phonon frequency comb, and there is no discernible impact on the comb's energy and spacing for magnetic fields up to 7T, suggesting that magnon excitations are not responsible for the emergence of frequency combs (also see Supplementary Fig. 1 for more details). This conclusion is supported by the fact that a 7T field would result in a 9.8 cm⁻¹ energy shift for magnons in this spin-3/2 system[28].

The second experiment examines how the energies of the frequency comb vary with temperature. Should magnon-phonon coupling influence this phonon mode, the temperature dependence would deviate from the expected behavior of the standard phonon-phonon scattering model, as reported in some $E_g$ modes in CrGeTe₃[24,29]. Figure 3 (and Supplementary Figs. 2 and 3) shows the detailed temperature dependence of the $A_g^5$ mode. From the color plot in Fig. 3a, b, we can see that the comb feature persists above $T_C$, with visible comb features up to temperatures higher than 150K for the 532 nm experiment. To systematically investigate the temperature dependence, we first fit the 532 nm Raman spectrum with a series of Voigt peaks with the same full width at half maximum (FWHM), and track their changes as a function of temperature (Fig. 3c). At least five peaks can be resolved in the fits below 150K, and four peaks can be

resolved above, although at above 200 K the peaks are significantly broadened and overlap together, making it less practical to distinguish them, which is consistent with the single broad peak observed in previous experiments in the literature[24,29]. Nevertheless, a temperature dependence of the peak energy and FWHMs can be extracted as plotted in Fig. 3d. Overall, the comb peaks soften and broaden with increasing temperature, which is consistent with the 3-phonon model. This result indicates that the phonon-phonon coupling will destroy the frequency comb by broadening the individual peaks.

To examine the potential influence of magnetic order on the $A_g^5$ phonon energy, we fit the phonon FWHMs with the 3-phonon scattering model:

$$\Gamma(T) = \Gamma(0) + \frac{b}{e^{\frac{E(0)}{2k_B T}} - 1}, \tag{1}$$

where $\Gamma(0)$ is the phonon FWHM at zero temperature, and $b$ is related to the scattering strength of the anharmonic decay[9,30,31]. From the fitted FWHM from Eq. (1), we can estimate the phonon frequency change $\Delta E(T) = E(0) - \sqrt{E(0)^2 - \Gamma(T)^2}$ from the damped oscillator model. The calculated $\Delta E(T)$ is around 0.03 cm⁻¹ at 300 K, which is significantly lower than the observed value of ~2.7 cm⁻¹. Therefore the softening of the phonon mode is not due to the anharmonic scattering. An alternative origin of the phonon softening is from thermal expansions, where the phonon energy is inversely proportional to the lattice unit volume in the first-order approximation. Since the temperature dependence of the cell volume has been determined[23], a fit can be made from equation $E(T) = E(0)[1 - \gamma \frac{V(T) - V(0)}{V(0)}]$ which connects the phonon frequency and thermal expansion. The mode-specific Grüneisen parameter $\gamma$ for the $A_g^5$ mode is obtained from this fit[32]. To obtain a meaningful estimate of the phonon energies and widths for the frequency comb, we compute a "weighted average" energy, $E_{avg}$, by summing the energy values of a given scan weighted by the corresponding Raman intensity subtracted by a linear background.

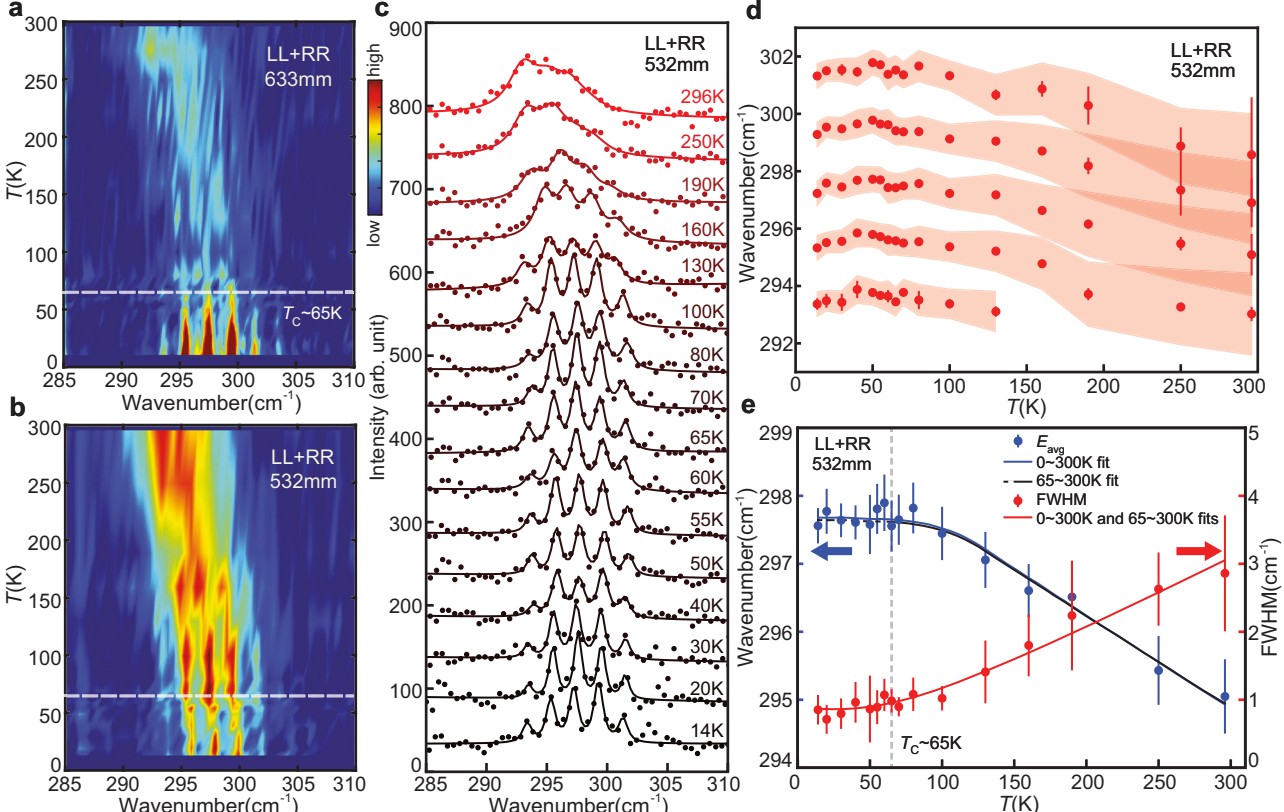

**Fig. 3 | Temperature dependence of phonon frequency combs in CrGeTe₃.**
**a**, **b** Temperature dependence of the $A_g^5$ mode, with 633 nm and 532 nm lasers respectively. **c** Peak profile fitting results at different temperatures of the 532 nm data. **d** Detailed temperature dependence of the peak positions (red dots) and FWHMs (red shading) of the fitting result in (**c**). **e** Fitting of the temperature dependence of the energy "weighted average" using thermal expansion data (Used

with permission of IOP Publishing, Ltd, from Crystallographic, magnetic and electronic structures of a new layered ferromagnetic compound $Cr_2Ge_2Te_6$. Carteaux et al.[23]; permission conveyed through Copyright Clearance Center, Inc.) and FWHMs using Eq. (1), the blue and red solid lines show the fitting with the whole dataset, while the black line shows the fitting with only $T \geq 65K$ data. Error bars in (**d**) and (**e**) represent one standard deviation.

We perform the fitting within the temperature ranges of [11, 300]K and [65, 300]K, yielding $E_0 = 297.68$ cm⁻¹ and 297.65 cm⁻¹, with $\gamma = 1.65$ and 1.62, respectively. These values are consistent with the macroscopic Grüneisen parameter (see Supplementary Information Section III), indicating that thermal expansion alone can account for the observed phonon softening. Additionally, both fits also yield identical $\Gamma_0 = 0.86$ cm⁻¹ and b = 2.31 cm⁻¹ (Fig. 3e), indicating that the temperature dependence of the anharmonic decay rate is not affected by magnetic order. These results suggest that the magnon-phonon coupling in this mode is not significant.

## Potential origin of phonon frequency combs

To understand the occurrence of the phonon frequency comb specifically at the $A_g^5$ mode of CrGeTe₃ and not in other modes or materials, we propose a phenomenological model after a careful analysis of this phonon mode. From previous ab initio calculations on the eigenmodes, this mode is characterized by the head-to-head motion of Ge dimers located at the center of the Cr hexagons, accompanied by the movement of Te atoms, while the Cr atoms surrounding these movements remain mostly stationary (Fig. 4a)[33]; Additionally, it can be revealed from the phonon energy calculations that this phonon branch has a very narrow, almost flat dispersion[25,26]. This indicates that the $A_g^5$ phonon mode is extremely localized, i.e., the oscillation of one $Ge_2Te_6$ cluster has minimal correlation effect on another. Since one phonon mode corresponds to only one degree of freedom, we can simplify this phonon mode into a collection of isolated quantum oscillators characterized by one variable $x$ which is the Ge-Ge distance relative to its equilibrium value (Fig. 4a). From the fact that typical optical and

phonon frequency comb generation requires nonlinearity, we assume a small, cubic nonlinear term in the harmonic oscillator Hamiltonian:

$$H = \frac{p^2}{2\mu} + \frac{1}{2}\mu\omega^2 x^2 + \lambda x^3, \tag{2}$$

where $p$ is the momentum of the mode, $\mu$ is an effective mass of the oscillator, and $\lambda$ is the cubic anharmonicity. Using second-order perturbation theory, we can calculate the energy difference between adjacent eigenstates $|n\rangle$ and $|n+1\rangle$ as:

$$E_{n+1} - E_n = \hbar\omega - \hbar A(n+1), \tag{3}$$

where $A$ is positive and is proportional to $\lambda^2$ (Fig. 4b). The subsequent step to generate frequency combs is to achieve a proper superposition of these states. To do this, we apply a semi-classical approach to the Raman scattering process. Specifically, we treat the oscillations within the phonon mode as purely quantum mechanical, whereas the interaction between atomic oscillations and the laser is regarded as classical. A suitable start for the analysis is to assume the laser can excite a coherent state of the oscillator, which is a classical analog in quantum oscillators[34,35]:

$$|\alpha_0\rangle = e^{-\frac{|\alpha_0|^2}{2}} \sum_{n=0}^{\infty} \frac{\alpha_0^n}{\sqrt{n!}} |n\rangle \tag{4}$$

where $\alpha_0$ is a complex eigenvalue for the coherent state, whose absolute value $|\alpha_0|$ is analogous to the amplitude for the coherent

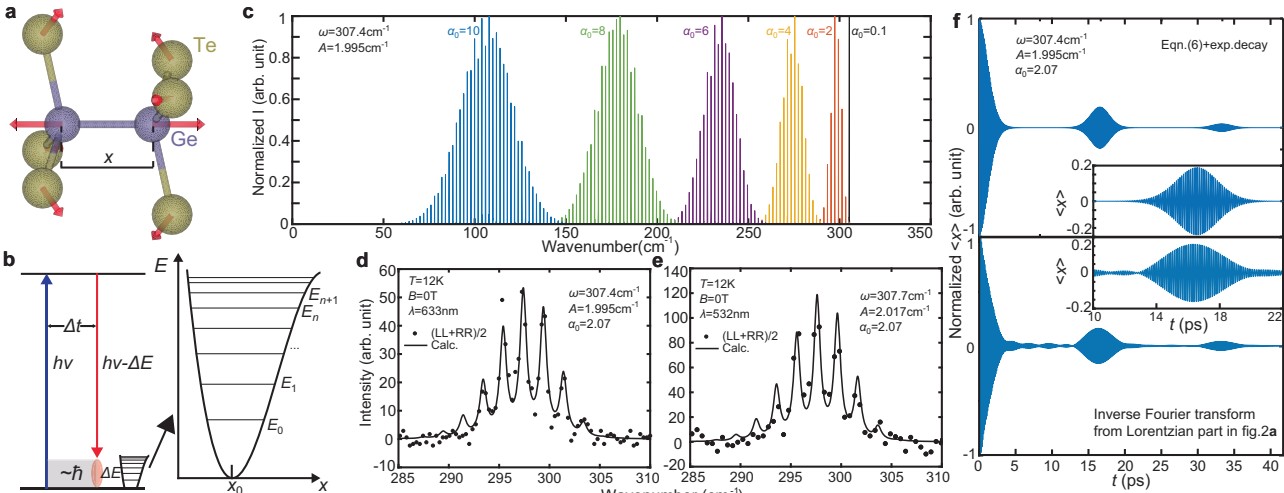

**Fig. 4 | Phenomenological model of phonon frequency combs. a** An illustration of the detailed oscillation pattern of the $A_g^5$ mode on the $Ge_2Te_6$ cluster. The arrows are not proportional to the actual atomic oscillation amplitude. **b** Schematics of the Raman scattering process, anharmonic potential, and associated energy levels. **c** Calculated frequency comb Raman spectra from Eq. (6) with different $\alpha_0$ values. **d**, **e** Fitting of the frequency comb with the Fourier transform of Eq. (6) for the 633 nm and 532 nm data, respectively. **f** Simulation of the atomic movement in the real space for the frequency comb in (**d**), compared with inverse Fourier transform from the Lorentzian part of the fit in Fig. 2a. The insets highlight the first revival of the oscillation.

state. In the oscillation process, every eigenstate $|n\rangle$ in the coherent state receives a phase factor $e^{-iE_n t/\hbar}$, and the time evolution of the expectation values can be calculated with the coherent state. Here we are interested in the expectation value of $x$:

$$\langle x \rangle_t = \langle a^\dagger + a \rangle_t = e^{-|\alpha_0|^2} \sum_{n=0}^{\infty} \frac{|\alpha_0|^{2n}\alpha_0}{n!} e^{-i(E_{n+1}-E_n)t/\hbar} + c.c. \quad (5)$$

where $a^\dagger$ and $a$ are the creation and annihilation operators respectively, and $c.c.$ stands for "complex conjugate". Putting $E_{n+1} - E_n = \hbar\omega - \hbar A(n+1)$ into the equation, we get:

$$\langle x \rangle_t = \alpha_0 \exp[-i(\omega - A)t + |\alpha_0|^2(e^{iAt} - 1)] + c.c. \quad (6)$$

Equation (6) describes the "classical" motion of atoms within the phonon mode, and a sample plot of this equation is illustrated in Fig. 4f. The Fourier transformation of Eq. (6), as shown in Fig. 4c, transforms the classical atomic movement of the phonon mode into its frequency domain, enabling a detailed analysis of the spectrum. In this spectrum, the parameter $\omega$ indicates the bare frequency of the oscillation, and the parameter $A$ is equal to the distance between adjacent spikes in the frequency comb. Meanwhile, $\alpha_0$ acts as a "window" that determines which part of the frequency comb is selected to appear in the spectrum. For small $\alpha_0$, the spectral weight is mainly located on the frequency $\omega - A$ which is the upper limit of the comb frequency. As $\alpha_0$ increases, the spectrum goes through a widening and a decrease in energy, indicating the nonlinear terms taking effect.

Consequently, the Raman intensity can be derived in a classical setting if only the expectation value of $x$ contributes to the Raman intensity[36]. The Raman process is based on the change in polarizability of the system as a function of the atomic displacement, which to the first order reads $\beta = \beta_0 + (\frac{\partial\beta}{\partial x})_0\langle x \rangle$ where $\beta$ is the polarizability and $\beta_0$ is its value at the equilibrium position. This connects the Raman intensity $I_R(\omega)$ with the atomic displacement (see Supplementary Information Section I) by $I_R(\omega) \propto (\langle x \rangle_\omega)^2$, where $\langle x \rangle_\omega$ is the Fourier transform of $\langle x \rangle_t$ into the frequency space. To fit the observed spectrum in $CrGeTe_3$, we utilize this model with the inter-spike distance $A = 1.995$ cm$^{-1}$ determined from prior analysis. From the simulation shown in Fig. 4c, the upper limit of the frequency comb is $\omega - A$, which may or may not be the #1 mode denoted in Fig. 2a. Therefore, we adjust both $\alpha_0$ and $\omega$ and apply the

Fourier transformation of Eq. (6) to simulate the spectrum. We fit the theoretical predictions with the observed data by convolving the spectrum with a Lorentzian. The optimal fitting parameters and the comparison between the calculated and experimental spectra are presented in Fig. 4d, e for the 633 nm and 532 nm scattering data respectively, showing that our model can well reproduce the observed frequency comb features. The fitting parameters suggest that the #1 mode is one spike below the upper limit of $\omega - A$, which is equal to the $E_0$ in the linearity fit shown in Fig. 2d. Furthermore, to compare the real space oscillation pattern between the theory and experiment, we calculate the inverse Fourier transform of the Lorentzian fits in Fig. 2a, and plot it with the calculation directly from Eq. (6) as shown in Fig. 4f. These two patterns closely resemble each other, suggesting that our model correctly captures the temporal evolution of the oscillations. Notably, the small linewidth of the observed frequency combs allows the signals to revive after more than 300 oscillations. When accounting for instrumental resolution, the lifetime of these oscillations extends to over 500 periods. This indicates that the anharmonicity parameter $A$ contributes minimally to phonon decay and primarily drives the parametric oscillations.

## Phonon frequency combs in $CrSiTe_3$

With the theory in mind, we conducted the same Raman scattering experiment on $CrSiTe_3$ which shares the same structure as $CrGeTe_3$, only with the Ge atoms replaced by Si[37]. This replacement has two effects: (1) it changes the atomic potential as well as anharmonicity, and (2) it alters the electron-phonon coupling in the $A_g^5$ mode, giving rise to a different driving amplitude. Figure 5 (and Supplementary Fig. 4) summarizes the Raman scattering results on the $A_g^5$ mode in $CrSiTe_3$[38,39]. Similar to $CrGeTe_3$, the $A_g^5$ mode in $CrSiTe_3$ has a flat dispersion at ~500 cm$^{-1}$[40], which is consistent with our observation of the main phonon peak at 518.4 cm$^{-1}$(Fig. 5a). Notably, two satellite peaks with weaker intensity are observed at 514.2 cm$^{-1}$ and 509.9 cm$^{-1}$, aligning with the model's prediction, as the frequency comb feature occurs only below the bare phonon frequency. A further simulation of the data with a smaller $\alpha_0$, and a larger $A$ can well reproduce the spectrum (Fig. 5a), indicating that the model from Eq. (6) has the ability to describe the frequency comb in both compounds, only with different parameters. Temperature dependence of the excitation is performed as well, revealing the survival of frequency comb up to 100 K

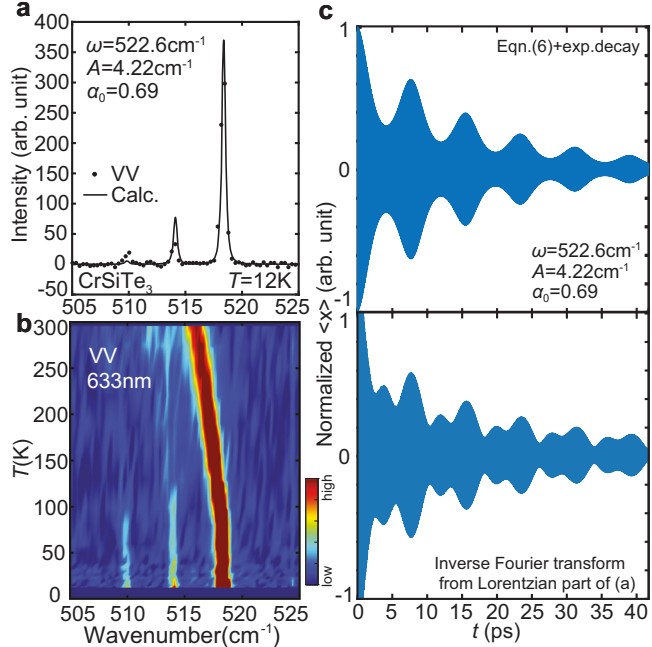

**Fig. 5 | Phonon frequency combs in CrSiTe₃, with parallel geometry (VV).**
**a** Raman spectrum of the $A_g^5$ mode in CrSiTe₃. The black solid line is a fit with the Fourier transform of Eq. (6). **b** Temperature dependence of the $A_g^5$ mode. **c** Simulation of the atomic movement for both theoretical and observed phonon frequency combs.

(Fig. 5b). The temporal evolution of the frequency combs in CrSiTe₃ features a weaker revival of the signal, consistent with the fact that smaller $\alpha_0$ represents less reveal of anharmonicity. Nevertheless, the signal still survives long time as reflected by the small linewidth observed in the Raman spectrum (Fig. 5c).

## Discussion

In this study, we have demonstrated the spontaneous formation of phonon frequency combs in the van der Waals materials CrGeTe₃ and CrSiTe₃ using high-resolution Raman scattering. Our simplified analysis, which includes anharmonic interactions within the atomic potential between Si/Ge dimers, matches the experimental results well. This suggests that the anharmonicity in these systems is not strongly linked to rapid phonon decay, a finding that raises interesting experimental and theoretical questions.

For example, the assumptions underlying our phenomenological model require further discussion. First, the origin of the coherent mode is not typically observed in spontaneous Raman scattering experiments, as higher-order excitations are significantly less probable due to the small coupling constant. A plausible explanation is that CrGeTe₃ is a narrow-gap semiconductor with a band gap of approximately 300 meV[41]. In this scenario, a Raman photon can excite an electron to a real state within the conduction band, with subsequent relaxation occurring via a cascade of phonon emissions, similar to processes observed in resonant Raman scattering[42]. This interpretation aligns with the fact that resonant excitation can enhance multiphonon processes, even when the individual phonon-photon coupling constants are weak. Moreover, the characteristic time scale of a typical Raman process is on the order of 10 fs. Based on the energy-time uncertainty relation, this corresponds to an energy uncertainty of approximately 500 cm⁻¹ (Fig. 4b). This energy uncertainty is consistent with the uncertainty associated with the coherent state in both CrGeTe₃ and CrSiTe₃, given by $\alpha_0\hbar\omega$.

The second issue concerning the assumption is that our model specifically requires that decoherence, as reflected in the

expectation value calculation, occurs during the phonon vibration process, instead of other Raman processes such as the absorption of the photon by the electrons and the detection of scattered light waves. This is a complex problem that demands both theoretical and experimental investigations beyond this work. On the theoretical side, a more detailed quantum model incorporating phonon, electron, and photon states is necessary to rigorously describe the interplay between these degrees of freedom and their role in decoherence. Experimentally, further studies could involve stimulated Raman scattering, which allows better control over the phonon oscillation amplitude[43].

Nevertheless, regardless of the theory details, we have shown experimentally that frequency-based technologies such as Raman scattering and infrared spectroscopy serves as a powerful probe for investigating high-frequency nonlinear phonon modes, particularly in the 5–15 THz range, where temporal techniques such as ultrafast pulsed lasers often face limitations in pulse bandwidth. Specifically, our work highlights Raman scattering as an effective tool for studying the dynamics of these modes and provides a route to explore frequency comb generation in optical phonon branches of crystalline solids, with implicit phonon coherence. To confirm this coherence, the development of new technologies that enable ultrafast phonon excitation in the 5–15 THz range will be necessary[44]. These findings offer insight into the generation and control of long-lived, tunable nonlinear phonons, with potential applications in ultrafast material control and phonon-based technologies.

## Methods

Single crystals of CrGeTe₃ were grown using a self-flux method. Cr, Ge, and Te powders in a mass ratio of 1:2:20 were mixed and sealed in a quartz tube under vacuum. The mixture was heated to 1050 °C and then cooled down to 475 °C. The crystals were harvested after removing the Ge-Te flux. CrGeTe₃ crystals were mounted in a closed-cycle helium cryostat (from Cryo Industries of America, Inc.) with a window for optical access. Surface layers of the crystals were exfoliated using adhesive tapes to obtain fresh sample surfaces. Single crystals of CrSiTe₃ were grown with a self-flux method using high-purity Cr, Si, and Te in a 1:2:6 molar ratio, loaded into a 5 ml alumina crucible. A catch crucible with quartz wool was placed above and the assembly sealed in a quartz tube under -1/3 atm of argon. The ampoule was heated to 1150 °C over 8 h, held for 16 h, then cooled to 700 °C at 3 °C/h, at which point the flux was decanted[40]. Raman measurements were conducted using both 532 nm and 633 nm lasers with 0.3 mW power. The laser beam was focused to a spot size of 2–3 µm on the probed sample using a 40x objective lens. Raman spectra were acquired using a Horiba LabRAM HR Evolution Raman microscope that is equipped with an 1800 grooves/mm grating and 100 µm slit width (with instrumental resolution of 0.4 cm⁻¹) and a thermoelectric-cooled CCD. A superconducting magnet (also from Cryo Industries of America, Inc.) was interfaced with the variable-temperature helium cryostat to apply an out-of-plane magnetic field. All measurements were performed at base pressure lower than $7 \times 10^{-7}$ torr.

## Data availability

The data supporting the findings of this study, including those used to generate the plots, are available on the Harvard Dataverse at https://doi.org/10.7910/DVN/HAT1SC. Part of the data in Fig. 3e and Supplementary Fig. S4b is adapted from refs. 23, 45–47, respectively.

## Code availability

The code used for plotting, fitting, and modeling the Raman spectra is available on the Harvard Dataverse at https://doi.org/10.7910/DVN/HAT1SC.

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

## Acknowledgements

L.C. and R.H. are grateful to Liuyan Zhao for fruitful discussions. The work at the University of California, Berkeley and Lawrence Berkeley National Laboratory was supported by the U.S. DOE under contract no. DE-AC02-05-CH11231 within the Quantum Materials Program (KC2202) (R.J.B.). R.H. acknowledges support by NSF Grants No. DMR-2300640 and DMR-2104036 and DOE Office of Science Grant No. DE-SC0020334 subaward S6535A. X.C.P. and G.C. would like to thank KAKENHI 22H00278 and 24K16998. D.G.M. and J.Y. acknowledge support from

the US Department of Energy, Office of Science, Basic Energy Sciences, Materials Science and Engineering Division. G.C., Y.P.C. and D.G.M. also acknowledge partial support from University of Southern California via Multidisciplinary University Research Initiatives (MURI) Program (Award No. FA9550-20-1-0322).

## Author contributions

L.C., R.J.B., and R.H. conceived and managed the project. The single-crystal $CrGeTe_3$ samples were grown by X.-C.P., K.T., G.C., Y.P.C., and A.E.L.A. The single-crystal $CrSiTe_3$ samples were grown by J.Y. and D.G.M. Raman scattering experiments were carried out by G.Y., C.N. under the supervision of R.H., and analyzed by L.C. Theoretical calculation is carried out by L.C. and N.G.-D. The paper was written by L.C., R.J.B., and R.H. with inputs from all coauthors.

## Competing interests

The authors declare no competing interests.
