## [Transparent Peer Review file · Nature Communications]

Spontaneously formed phonon frequency combs in van der Waals solid CrGeTe₃ and CrSiTe₃

Corresponding Author: Dr Lebing Chen

Version 0:

Reviewer comments:

Reviewer #1

(Remarks to the Author)

Report on "Spontaneously formed phonon frequency combs in van der Waals solid CrXTe₃ (X=Ge,Si)" by Chen et al.

Overall this is a very interesting paper dealing with phonon frequency combs observed in Raman scattering of van der Waals materials.

In general, I would recommend publication provided that the remaining issues are resolved by the authors.

Anharmonic decay fitting:

There seems to be an obvious discrepancy between the experimental data shown in Fig. 3(a) to (c) and Fig. 3 (d). Clearly a broad mode is splitting into 6 sharp comb modes. The temperature dependence shows only one mode where one would have expected 6 modes splitting at a certain temperature above T_c. This also raises the question if a simple anharmonic dependence can be extracted. The anharmonic behavior would also reflect in a simultaneous hardening and sharpening that can be fitted within one model. So the dependence of the widths of the phonons would need to be discussed as well. The error bars in Fig. 3(d) are small enough to see the evolution of 6 peaks from the bulk of one broad peak at 300 K. However, how can the error bar at 300 K only be 0.5 to 1 cm⁻¹ when there is no clear peak structure anymore? At the moment this Fig. 3(d) seems to be contradicting the main claim and needs to be corrected. Fig. S4 and S5 try to address this issue, but they seem to be inconsistent with Fig. 3(d) in the same temperature range. Further, Fig. S4 / S5 show the widths and frequencies of the 6 modes only below 100 K. However, most of the anharmonic decay happens between 60 and 300 K and, therefore, improvements on Fig. 3 and its discussion are mandatory.

In general, I think the authors correctly point out that anharmonic couplings are responsible for the vanishing of the frequency combs at higher temperatures. However, I find the current presentation of the fits of the data not convincing.

A quantum model:

Let us be honest. The authors do not consider any real light-matter interaction at all (at no point in the text we can find a vector potential of an electromagnetic field). We are just discussing an anharmonic oscillator that is not coupled to anything. Thus, sentences suggesting other wise (semiclassical and classical light matter interaction etc.) can simply be eliminated.

Then, the derived parameter 'A' should be related to the model of the anharmonic decay as it is a measure on anharmonicity.

However, what I do not understand is the following: For weak anharmonic coupling at T = 0 all excitations should be in the harmonic limit (i.e. close to the bottom of the parabola) and all phonons would occupy n=0 (i.e. the ground state). The excitation of a phonon would then always have the same energy (n=0 to n=1). Only higher excited state excitations would shift in energy depending on the strength of A to value slowly decreasing in energy as n increases (See Fig. 4 (b)). For this to happen in a one-phonon process higher energy states would need to be occupied. The authors show no reason for this to happen.

This, however, raises the question how the two-phonon spectrum looks for the Ag5 mode and especially how the power dependence (fluence) looks like.

In general, one would rather calculate the phonon Greens function and relate it to the imaginary part of the Raman response. This is then related to the scattering cross section.

It looks to me that the model proposed by the authors cannot work as the “response” is not related to light scattering, but rather to an amplitude of the phonon vibration. Its connection to the Raman response for spontaneous Raman scattering is well worked out in the literature, but somehow missed by the authors.

In general, the authors suggest the presence of a “coherent state” in a linear spontaneous Raman scattering experiment. “Coherent” Raman states typically develop in two or three photon beam experiments such as stimulated Raman or CARS where the sample acts as a “gain medium”. Such an experiment clearly has not been performed. The large amplitudes required for the anharmonic term to have an effect would need a second laser driving the system at the phonon frequency – see (e.g. <https://journals.aps.org/prb/abstract/10.1103/PhysRevB.108.075419>) being an MIR-pump spontaneous Raman probe experiment which is only possible with an IR mode being driven by a THz laser (see also Ref. 1 to 4) and a Raman mode as spectator.

In summary, the authors have very interesting data that clearly deserve publication, but the level of inconsistencies appear to be too significant. The current version cannot be published, but I am optimistic that a revised version can meet the standards of Nature Communications.

Minor comments: Units have sometime spaces to their values sometimes not. In text Fig. X is sometimes written in capital letters some not (Fig X versus fig X). Raman laser -> laser or incident photons (there is only one laser in this experiment).

Reviewer #2

(Remarks to the Author)

The authors report the observation of long-lived nonlinear optical phonons through the spontaneous formation of phonon frequency combs in the van der Waals material CrXTe₃ (X = Ge, Si), as revealed by high-resolution Raman scattering. Unlike conventional optical phonons, the highest Ag mode in CrGeTe₃ was found to split into sharp, equidistant peaks, forming a frequency comb that remains stable for hundreds of oscillations and persists up to 100 K before decaying. This work deserves attention as it demonstrates the effectiveness of Raman scattering in probing high-frequency nonlinear phonon modes, offering valuable insights into the generation of long-lived, tunable phonon frequency combs.

Before considering publication, a few concerns should be addressed:

1. The authors claim an impressive spectral resolution of 0.4 cm⁻¹ for a single-stage spectrometer. Experimental evidence supporting this claim should be provided.
2. Why did the authors not analyze the line profiles using Voigt profiles, given their relevance to accurately describing line shapes?
3. While the authors assert that anharmonic effects are weak for the highest Ag mode, why was the impact of thermal expansion on the temperature dependence of the phonon energy not considered?

Reviewer #3

(Remarks to the Author)

This is a very clear paper demonstrating in a highly convincing manner the observation of phonon combs in the van der Waals materials CrGeTe₃ and CrSiTe₃. The clarity of the presentation and thoroughness of the work (including the convincing demonstration that the observed combs are due to phonons rather than magnons) do not leave room for many questions. Nevertheless, I would appreciate if the authors could elaborate on the following:

- 1) I do not understand Eq. (3). It seems that the energy difference between adjacent phonon states increase indefinitely with n. This is in contrast to figure 4b, which suggests that the energy difference should decrease with n.
- 2) In the standard interpretation/description of Raman spectroscopy the photon energy loss is due to excitation of phonons. If only first order processes are considered then only single phonons are excited meaning that Raman peaks will correspond energy differences between phonon states (n and n+1) with weights given by the thermal occupation of the phonons (Bose Einstein distribution for thermal equilibrium). What would be the result if a thermal distribution of phonons is assumed rather than a coherent state (Eq. (4))?
- 3) To follow up on the previous question: What is the motivation for assuming that the phonon state is a coherent state? Is that choice made to make analytical calculations possible? Is it due to the fact that the coherent state describes the exciting laser field?
- 4) Related to the previous: Can one show that a coherent laser will prepare the phonons in a coherent state (if the phonon coherence time is sufficiently long)?

Version 1:

Reviewer comments:

Reviewer #1

(Remarks to the Author)

The authors have addressed many of my comments. In particular inconsistencies are removed from the current manuscript and additional information has been provided. The reader can now judge the quality of the manuscript much better. I still think that the calculation of the Raman cross section is simplified but I think this is clearly spelled out. I am convinced that the manuscript will stimulate the scientific community and it provides a very interesting result. I therefore strongly recommend the publication of the current version of the manuscript.

Reviewer #2

(Remarks to the Author)

I am not fully satisfied with all the replies to my previous comments. First, the Authors should now be aware that the spectral resolution strongly depends on the entrance slit width. This parameter must be explicitly stated, especially since the claimed performance (in terms of resolution) lies at or near the limits of what the experimental setup can realistically achieve.

Furthermore, as I previously pointed out, I do not find the model described by Eq. (1) to be satisfactory—particularly regarding the temperature dependence of the energy. For an alternative approach, see the discussion in Phys. Rev. B 90, 024411 (2014).

Overall, the revised manuscript has been improved; however, the Authors still need to address these issues in more detail before I can recommend it for publication.

Reviewer #3

(Remarks to the Author)

The authors have replied satisfactorily to all the questions in my initial report and the manuscript can now be published.

Version 2:

Reviewer comments:

Reviewer #2

(Remarks to the Author)

The authors have made all the necessary corrections. I recommend the manuscript for publication.

made.

Reviewer #1 (Remarks to the Author):

Report on “Spontaneously formed phonon frequency combs in van der Waals solid CrXTe₃ (X=Ge,Si)” by Chen et al.

Overall this is a very interesting paper dealing with phonon frequency combs observed in Raman scattering of van der Waals materials.

In general, I would recommend publication provided that the remaining issues are resolved by the authors.

Reply: We sincerely appreciate the referee’s positive assessment of our work and their recommendation for publication. We are grateful for the constructive feedback and will carefully address the remaining issues to improve the manuscript.

Anharmonic decay fitting:

There seems to be an obvious discrepancy between the experimental data shown in Fig. 3(a) to (c) and Fig. 3 (d). Clearly a broad mode is splitting into 6 sharp comb modes. The temperature dependence shows only one mode where one would have expected 6 modes splitting at a certain temperature above T_c.

Reply: We thank the referee for pointing this out. Previously in the main text, we used a single Gaussian fit for the low-temperature data to extract a ‘weighted average’ of the peak center. However, we acknowledge that showing only a single-peak fit may be misleading given the clear presence of six visible peaks. To address this, we have incorporated the sub- peak fitting, previously shown in Fig. S5(a), into the main figure, with a more systematic fitting result. We also use Raman intensity as the weight to get a ‘weighted average’ of the peak center for the anharmonic model fitting, shown in the revised fig.3.

This also raises the question if a simple anharmonic dependence can be extracted. The anharmonic behavior would also reflect in a simultaneous hardening and sharpening that can be fitted within one model. So the dependence of the widths of the phonons would need to be discussed as well.

Reply: We agree that the peak widths are important in anharmonic fittings, and we have revised fig.3 to include the widths, and fit them with eqn.1 in fig.3e. The width increases with temperature, which is consistent with the anharmonic decay model. However, at higher temperatures above 200K, resolving individual peak widths becomes increasingly difficult as they broaden and overlap, making it challenging to distinguish between them.

The error bars in Fig. 3(d) are small enough to see the evolution of 6 peaks from the bulk of one broad peak at 300 K. However, how can the error bar at 300 K only be 0.5 to 1 cm^{-1} when there is no clear peak structure anymore?

Reply: The error bars we showed are obtained from fitting the peaks with Gaussian function in Matlab. Each Gaussian peak has three parameters: amplitude, peak center, and peak width. Each of these three parameters has its own error given by Matlab fitting. The error bars we show in Fig. 3 are the fitting errors of the peak frequency, and thus are much smaller than the peak width.

At the moment this Fig. 3(d) seems to be contradicting the main claim and needs to be corrected. Fig. S4 and S5 try to address this issue, but they seem to be inconsistent with Fig. 3(d) in the same temperature range.

Reply: To address this issue, we have replaced fig.3 with a new one, focusing more on the temperature dependence of the individual peaks.

Further, Fig. S4 / S5 show the widths and frequencies of the 6 modes only below 100 K. However, most of the anharmonic decay happens between 60 and 300 K and, therefore, improvements on Fig. 3 and its discussion are mandatory.

From equation (1) one can make a reasonable estimate of the decay rate at different temperatures. For a phonon excitation at $\sim 300\text{cm}^{-1}$, the ratio of the decay rate at 300K vs 100K is about 4:1 (For 300K to 60K it is 16:1), which is consistent with the referee's remark.

To further solidify the dataset, we also extract the temperature dependence of the phonon energy and linewidth of CrSiTe_3 since its comb structure is less pronounced with almost no overlap. Both softening and broadenings are observed throughout the whole temperature range, also with decay happening mostly at higher temperatures (See Supplemental fig.S4).

In general, I think the authors correctly point out that anharmonic couplings are responsible for the vanishing of the frequency combs at higher temperatures. However, I find the current presentation of the fits of the data not convincing.

In response of the reviewer's comments, we have made the following changes to this section: 1. We have completely replaced Fig. 3 with a new figure that highlights the temperature dependence of the frequency comb spike energy and width. The previous one- Gaussian fit has been moved to Fig. S2 in the Supplementary Information. 2. We have

incorporated the temperature dependence of the width into Equation 1 and fitted the experimental data accordingly. 3. We have updated the main text to reflect these changes.

A quantum model:

Let us be honest. The authors do not consider any real light-matter interaction at all (at no point in the text we can find a vector potential of an electromagnetic field). We are just discussing an anharmonic oscillator that is not coupled to anything. Thus, sentences suggesting other wise (semiclassical and classical light matter interaction etc.) can simply be eliminated.

We agree with the reviewer that our discussion of light scattering was insufficient. Our initial presumption is that the final energy-resolved detection of Raman laser is a classical process, therefore an expectation value must be calculated at some point during the process. Furthermore, as the Raman scattered amplitude is directly proportional to the polarizability change of the sample, which is then proportional to the atomic displacement x , therefore we chose to calculate the expectation value of x , with the square of its Fourier transformation to the frequency space proportional to the Raman intensity. We have added these relations to the main text, with detailed deductions in the Supplemental Information.

Although we agree that a proper light-matter interaction calculation will involve vector potential or electromagnetic fields, we find it unnecessary to dive into such details. In this experiment, the scattering intensity is polarization-independent because the Ag5 mode oscillation direction is parallel to both the incident and scattered light. As shown in Fig. R3, none of the Raman-active modes exhibit significant angle dependence in their scattering intensity. Therefore, a photonic description based solely on frequency and an electric field without specified directions is sufficient in this case.

VV - Polar Plots for Each Component

Fig. R1: Angle dependence of CrGeTe_3 Raman modes at 11 K. The laser is in linearly parallel polarization configuration. No obvious angle dependence is observed.

Then, the derived parameter 'A' should be related to the model of the anharmonic decay as it is a measure on anharmonicity.

This question is important, as we are actually talking about two different kinds of anharmonicity here.

The anharmonicity related to frequency comb generation is within the Ag5 phonon mode, which corresponds to the energy difference between adjacent n-phonon modes.

In contrast, the anharmonic decay model describes the anharmonic coupling between the Ag5 phonon mode to other phonon modes, where the Ag5 phonon at $q=0$ either decays into or is effectively "synthesized" by two other phonon modes with $+q$ and $-q$ in momentum.

This part of anharmonicity contributes mostly to the anharmonic decay observed in fig.3.

These two anharmonicities can have different strengths, and the temperature dependence is more pronounced in the latter because it involves lower energy phonons. Therefore, these two parameters are not necessarily connected.

However, what I do not understand is the following: For weak anharmonic coupling at $T = 0$

all excitations should be in the harmonic limit (i.e. close to the bottom of the parabola) and all phonons would occupy $n=0$ (i.e. the ground state). The excitation of a phonon would then always have the same energy ($n=0$ to $n=1$). Only higher excited state excitations would shift in energy depending on the strength of A to a value slowly decreasing in energy as n increases (See Fig. 4 (b)). For this to happen in a one-phonon process higher energy states would need to be occupied. The authors show no reason for this to happen.

We greatly appreciate the referee for pointing this out. Here we give a heuristic way to understand why there is a possibility to excite higher n at the ground state, we can review the Raman scattering process. A Raman scattering process includes (1) the absorption of a photon to a virtual high-energy state, and (2) the decay of the high-energy state into a scattered photon and a phonon state. Since the Raman laser has photon energy in eV, it is possible to decay into higher energy states than $n=0$. Notably, the whole absorption-decay process takes place in a time scale on the order of 10 fs, and due to the energy-time uncertainty relation, the corresponding energy uncertainty is $\sim 500\text{cm}^{-1}$ if we assume $\Delta t \Delta E \sim \hbar$, whereas our energy uncertainty of the proposed coherent state of $\alpha_0 = 2$ in CrGeTe₃ is $\alpha \hbar \omega \sim 600\text{cm}^{-1}$.

This, however, raises the question how the two-phonon spectrum looks for the Ag₅ mode and especially how the power dependence (fluence) looks like.

We agree that it is indeed very important to measure the two-phonon spectrum as well as the power dependence, therefore we have measured them as plotted in fig.R2. To summarize, the Raman spectrum at 600cm^{-1} shows no sign of two-phonon mode, and the power does not seriously affect the frequency comb structure, except for the high power of 2.1W potentially heating the sample and resulting in the softening and broadening of the peak.

The absence of the two-phonon mode is somewhat unexpected as the one-phonon mode has flat dispersion, which makes the two-phonon density of states extremely focused on an energy at around 600cm^{-1} . We are not entirely sure about the reason of this absence, but one explanation is that the two-phonon mode's wavefunction is already calculated in the coherent state and contributes to the frequency comb instead of the two-phonon mode intensities. Of course, this can also be simply because two-phonon excitations are a higher-order effect.

The power/fluence independence provides evidence that the frequency comb generation in spontaneous Raman scattering is not a stimulated effect, as the comb structure will otherwise be changed due to the change in the amplitude of the driving force. Therefore, it

gives indirect evidence that the frequency comb generation here is a one-photon process, and the coherent state is from the energy-time uncertainty relations.

Fig. R2: Left: Raman intensity at the two-phonon energy ($\sim 590\text{cm}^{-1}$) for the Ag5 mode. The spike at around 593cm^{-1} is an artifact. Right: Power dependence of the Ag5 mode.

In general, one would rather calculate the phonon Greens function and relate it to the imaginary part of the Raman response. This is then related to the scattering cross section.

It is surely a standard way to calculate the phonon Green's function and use its imaginary part as the spectral function. Nevertheless, we do not think it is necessary because of the following reasons: First, in the current case, both the Raman process and the phonon excitations are not q -dependent, as the phonon has a flat dispersion (Fig. R3), and Raman only probes excitations at the Gamma point. Therefore, quantum mechanical treatments with only frequency dependence will suffice. Second, the spectral function is an indication of viable density of states at a certain frequency and wavevector, while it is obvious that no density of states is available at the frequency comb peaks if we only use the energy eigenstates (Fock states) for calculation, and therefore we cannot obtain a reasonable spectrum of frequency combs as an explanation.

There are other problems for using the Greens function on a coherent state, such as the indeterministic excitation energy (which makes the Lehmann representation hard to apply), the state coherence on perturbation, etc. We believe that this is beyond the scope of this work and worth another dedicated piece of work to explain.

Fig. R3: Calculated phonon dispersion in CrGeTe₃ from Refs. [25] (Reprinted figure with permission from Zhang, B. H., Hou, Y. S., Wang, Z. & Wu, R. Q. First-principles studies of spin-phonon coupling in monolayer Cr₂Ge₂Te₆. Phys. Rev. B 100, 224427 (2019). Copyright (2019) by the American Physical Society.) and [27] (adapted from Chen, L., Mao, C., Chung, JH. et al. Anisotropic magnon damping by zero-temperature quantum fluctuations in ferromagnetic CrGeTe₃. Nat Commun 13, 4037 (2022). <https://doi.org/10.1038/s41467-022-31612-w>) from the main text. The highest energy mode is the Ag5 mode.

It looks to me that the model proposed by the authors cannot work as the “response” is not related to light scattering, but rather to an amplitude of the phonon vibration. Its connection to the Raman response for spontaneous Raman scattering is well worked out in the literature, but somehow missed by the authors.

We have added a detailed discussion between the amplitude and Raman response in the main text as well as the supplemental information. Furthermore, we have refined our claim, replacing the term ‘quantum model’ with ‘phenomenological model,’ as a true quantum model would require a much more detailed analysis of photon, electron, and lattice interactions.

In general, the authors suggest the presence of a “coherent state” in a linear spontaneous Raman scattering experiment. “Coherent” Raman states typically develop in two or three photon beam experiments such as stimulated Raman or CARS where the sample acts as a “gain medium”. Such an experiment clearly has not been performed. The large amplitudes required for the anharmonic term to have an effect would need a second laser driving the system at the phonon frequency – see (e.g. <https://journals.aps.org/prb/abstract/10.1103/PhysRevB.108.075419>) being an MIR- pump spontaneous Raman probe experiment which is only possible with an IR mode being driven by a THz laser (see also Ref. 1 to 4) and a Raman mode as spectator.

We also think that it is strange to observe such a coherent state in a spontaneous Raman scattering experiment, and we admit that we do not have a better

understanding of the origin of the coherent state that goes deeper than the energy-time uncertainty relations in the Raman process. On the other hand, the referee has enlightened us with potential future experiments on alternative ways to control and generate frequency combs in solid-state systems, and shed more light to future directions from this experiment. We have mentioned this and added the PRB reference in the discussion section.

In summary, the authors have very interesting data that clearly deserve publication, but the level of inconsistencies appear to be too significant. The current version cannot be published, but I am optimistic that a revised version can meet the standards of Nature Communications.

We sincerely thank the reviewer again for giving such thoughtful and enlightening comments. We have rewritten the part the referee mentioned, especially the discussion on the potential origin of the phonon frequency combs, with more focus on the light-matter interactions as well as the potential source of the coherent state. We hope our revision can meet the reviewer's expectations.

Minor comments: Units have sometime spaces to their values sometimes not. In text Fig. X is sometimes written in capital letters some not (Fig X versus fig X). Raman laser -> laser or incident photons (there is only one laser in this experiment).

We have fixed these issues pointed out by the reviewer.

Reviewer #2 (Remarks to the Author):

The authors report the observation of long-lived nonlinear optical phonons through the spontaneous formation of phonon frequency combs in the van der Waals material CrXTe₃ (X = Ge, Si), as revealed by high-resolution Raman scattering. Unlike conventional optical phonons, the highest Ag mode in CrGeTe₃ was found to split into sharp, equidistant peaks, forming a frequency comb that remains stable for hundreds of oscillations and persists up to 100 K before decaying. This work deserves attention as it demonstrates the effectiveness of Raman scattering in probing high-frequency nonlinear phonon modes, offering valuable insights into the generation of long-lived, tunable phonon frequency combs.

We thank the reviewer for his/her positive evaluation of our work.

Before considering publication, a few concerns should be addressed:

1. The authors claim an impressive spectral resolution of 0.4 cm^{-1} for a single-stage spectrometer. Experimental evidence supporting this claim should be provided.

We thank the reviewer for raising this question. We used a commercial Horiba LabRAM HR Evolution Raman Microscope system to take our Raman data. The Raman spectrometer uses an 800 mm focal length astigmatic flat field spectrograph based on concave mirrors, providing a very high spectral resolution. An example of an original data set taken using the Horiba LabSpec software is shown in the table below. The first column is the Raman shift in cm^{-1} , and the second column is the Raman intensity in counts. We can see that the Raman frequency difference between each CCD pixel (the difference between adjacent values in the first column) is $\sim 0.35 \text{ cm}^{-1}$, which is rounded to $\sim 0.4 \text{ cm}^{-1}$. Furthermore, as shown in Fig.S4, the lowest FWHM probed in this work is $\sim 0.43 \text{ cm}^{-1}$ in the CrSiTe3 Ag5 mode, which provides an upper limit of the resolution. This is the reason for us to claim a spectral resolution of 0.4 cm^{-1} .

0.04566	64555
0.39164	64555
0.73608	64555
1.08203	11305
1.42643	726
1.77235	421
2.11673	277
2.46261	243
2.80696	204
3.15282	175
3.49713	210
3.84296	242
4.18725	264
4.53304	300
4.8773	325
5.22307	327
5.56729	356
5.91151	352
6.25723	745
6.60141	3627
6.9471	4139
7.29125	1214
7.63691	371
7.98103	98
8.32514	71
8.67075	72
9.01483	60

2. Why did the authors not analyze the line profiles using Voigt profiles, given their relevance to accurately describing line shapes?

We thank the reviewer for this suggestion. It is indeed better to fit the line profiles with Voigt functions, and we have re-fitted all relevant lines with Voigt again for accuracy. Additionally,

since the Voigt profile introduces more parameters into the fitting, we have fixed the Gaussian component to the instrumental resolution to ensure a reliable fit.

3. While the authors assert that anharmonic effects are weak for the highest Ag mode, why was the impact of thermal expansion on the temperature dependence of the phonon energy not considered?

We believe that the referee is asking on the anharmonicity's effect on the softening of the phonon mode with rising temperature. While it is indeed observed as in Fig.3 that softening is visible, we do not think that is directly related to the anharmonicity shown in Equation (2).

There are two types of anharmonicity associated with this phonon mode: the first is shown in equation (2) which contributes to the energy difference between adjacent n -phonon modes, and the second is shown in equation (1) corresponding to the coupling between Ag5 and other lower energy phonon modes, which is the main source of phonon softening. Both of them are weak as the comb distance as well as the softening is small compared to the bare phonon energy.

Reviewer #3 (Remarks to the Author):

This is a very clear paper demonstrating in a highly convincing manner the observation of phonon combs in the van der Waals materials CrGeTe₃ and CrSiTe₃. The clarity of the presentation and thoroughness of the work (including the convincing demonstration that the observed combs are due to phonons rather than magnons) do not leave room for many questions.

We appreciate the reviewer's positive feedback on our work.

Nevertheless, I would appreciate if the authors could elaborate on the following:

1) I do not understand Eq. (3). It seems that the energy difference between adjacent phonon states increase indefinitely with n . This is in contrast to figure 4b, which suggests that the energy difference should decrease with n .

We thank the reviewer for asking this question. Eq. (3) left-hand side is the energy difference between adjacent states. On the right-hand side, we can see that as n increases, the energy difference decreases, which is consistent with Fig. 4b, as pointed out by the reviewer.

2) In the standard interpretation/description of Raman spectroscopy the photon energy loss is due to excitation of phonons. If only first order processes are considered then only single phonons are excited meaning that Raman peaks will correspond energy differences between phonon states (n and $n+1$) with weights given by the thermal occupation of the phonons (Bose Einstein distribution for thermal equilibrium). What would be the result if a thermal distribution of phonons is assumed rather than a coherent state (Eq. (4))?

This is a good question as it displays the impossibility to consider these phonon frequency combs as a thermal effect. From Bose-Einstein distribution, we can know that the population of phonons at state n follows the Bose factor $1/[\exp(E_n/k_B T)-1]$. For $E_1 \sim 300 \text{cm}^{-1} \sim 37 \text{meV}$ at 11K, the Bose factor will be on the order of 10^{-17} (for higher order excitations this can only be exponentially worse), which is impossible to measure.

3) To follow up on the previous question: What is the motivation for assuming that the phonon state is a coherent state? Is that choice made to make analytical calculations possible? Is it due to the fact that the coherent state describes the exciting laser field?

There are a few motivations for assuming a coherent state. The first reason is because we observed more phonon peaks than symmetry allowed, and we have to introduce new degrees of freedom to account for the frequency comb peaks observed near Ag5, which leads us to assume a superposition state in the first place. Second, our knowledge of classical light-matter interactions is mostly based on a coherent state description, for example, in the classical theory of Raman scattering, the Raman intensity is proportional to the square of phonon atomic displacement, and calculating the expectation value of $\langle x \rangle$ from a coherent state is a good way to achieve a classical atomic displacement.

4) Related to the previous: Can one show that a coherent laser will prepare the phonons in a coherent state (if the phonon coherence time is sufficiently long)?

At the current understanding, we do not have evidence that either proves or disproves that a coherent laser will prepare coherent phonons, which also makes this an intriguing question as it clears the map for future experiments such as stimulated Raman scattering to generate and tune the phonon frequency comb in various ways. We have made a remark of this in the discussion section.

Reviewer #1 (Remarks to the Author):

The authors have addressed many of my comments. In particular inconsistencies are removed from the current manuscript and additional information has been provided. The reader can now judge the quality of the manuscript much better. I still think that the calculation of the Raman cross section is simplified but I think this is clearly spelled out. I am convinced that the manuscript will stimulate the scientific community and it provides a very interesting result. I therefore strongly recommend the publication of the current version of the manuscript.

Reply: We thank the reviewer very much for the positive feedback and thoughtful comments. We are glad to hear that our revisions have improved the clarity and consistency of the manuscript, and we appreciate the reviewer's support for publication.

Reviewer #2 (Remarks to the Author):

I am not fully satisfied with all the replies to my previous comments. First, the Authors should now be aware that the spectral resolution strongly depends on the entrance slit width. This parameter must be explicitly stated, especially since the claimed performance (in terms of resolution) lies at or near the limits of what the experimental setup can realistically achieve.

Reply: We thank the reviewer for raising this question. The spectral resolution does depend on the entrance slit width. The slit width used in our experiments is $100\ \mu\text{m}$. We used this slit width because it gives sufficient resolution for us to resolve the frequency combs and sufficient signal intensities so that the spectra have good signal-to-noise ratio. We have added this slit width information in the 'Method' section of the manuscript.

Furthermore, as I previously pointed out, I do not find the model described by Eq. (1) to be satisfactory—particularly regarding the temperature dependence of the energy. For an alternative approach, see the discussion in Phys. Rev. B 90, 024411 (2014).

Reply: We sincerely appreciate the reviewer's guidance and the helpful reference. After carefully studying the suggested work (Phys. Rev. B 90, 024411), we recognize that our original anharmonic phonon decay model alone is insufficient to fully explain the observed temperature-dependent phonon softening. In response, we have revisited our analysis by incorporating both anharmonic decay and thermal expansion effects. Our revised model shows that thermal expansion plays a dominant role in the softening of this particular phonon mode compared to anharmonic interactions. We have added a detailed discussion

of this revised analysis in the Supplemental Materials and have also addressed the issue in the main text.

Overall, the revised manuscript has been improved; however, the Authors still need to address these issues in more detail before I can recommend it for publication.

Reply: We thank the reviewer for his/her continued evaluation and constructive feedback. We have addressed the remaining issues in more detail to further improve the manuscript. We hope that these revisions meet the reviewer's expectations.

Reviewer #3 (Remarks to the Author):

The authors have replied satisfactorily to all the questions in my initial report and the manuscript can now be published.

Reply: We sincerely appreciate the reviewer's positive assessment and support for publication.